# Characterization of the Secretome of a Specific Cell Expressing Mutant Methionyl-tRNA Synthetase in Co-Culture Using Click Chemistry

**DOI:** 10.3390/ijms23126527

**Published:** 2022-06-10

**Authors:** Sungho Shin, Seonjeong Lee, Sunyoung Choi, Narae Park, Yumi Kwon, Jaehoon Jeong, Shinyeong Ju, Yunsil Chang, Kangsik Park, Chulwon Ha, Cheolju Lee

**Affiliations:** 1Chemical & Biological Integrative Research Center, Korea Institute of Science and Technology, Seoul 02792, Korea; sungho@kist.re.kr (S.S.); sejelee@kist.re.kr (S.L.); nr_park@kist.re.kr (N.P.); ymkwonlab@gmail.com (Y.K.); syju@kist.re.kr (S.J.); 2KHU-KIST Department of Converging Science and Technology, Graduate School, Kyung Hee University, Seoul 02447, Korea; kspark@khu.ac.kr; 3Division of Bio-Medical Science and Technology, KIST School, Korea University of Science and Technology, Seoul 02792, Korea; 4Department of Orthopedic Surgery, Samsung Medical Center, School of Medicine, Sungkyunkwan University, Seoul 06351, Korea; jsm002love@nate.com (S.C.); chulwon.ha@gmail.com (C.H.); 5Cell and Gene Therapy Institute, Samsung Medical Center, Seoul 06351, Korea; yunsil.chang@gmail.com; 6Department of Health Sciences and Technology, SAIHST, Sungkyunkwan University, Seoul 06351, Korea; 7Division of Radiation Biomedical Research, Korea Institute of Radiological and Medical Sciences, Seoul 01812, Korea; jeongj@kirams.re.kr; 8Department of Pediatrics, Samsung Medical Center, School of Medicine, Sungkyunkwan University, Seoul 06351, Korea; 9Department of Physiology, School of Medicine, Kyung Hee University, Seoul 02447, Korea

**Keywords:** co-culture, secretome, azidonorleucine, click chemistry, BONCAT, mesenchymal stromal cells

## Abstract

Co-culture system, in which two or more distinct cell types are cultured together, is advantageous in that it can mimic the environment of the in vivo niche of the cells. In this study, we presented a strategy to analyze the secretome of a specific cell type under the co-culture condition in serum-supplemented media. For the cell-specific secretome analysis, we expressed the mouse mutant methionyl-tRNA synthetase for the incorporation of the non-canonical amino acid, azidonorleucine into the newly synthesized proteins in cells of which the secretome is targeted. The azidonorleucine-tagged secretome could be enriched, based on click chemistry, and distinguished from any other contaminating proteins, either from the cell culture media or the other cells co-cultured with the cells of interest. In order to have more reliable true-positive identifications of cell-specific secretory bodies, we established criteria to exclude any identified human peptide matched to bovine proteins. As a result, we identified a maximum of 719 secreted proteins in the secretome analysis under this co-culture condition. Last, we applied this platform to profile the secretome of mesenchymal stem cells and predicted its therapeutic potential on osteoarthritis based on secretome analysis.

## 1. Introduction

Mesenchymal stem cells (MSCs) are pluripotent adult stem cells, which are found not only in Wharton’s jelly and placenta but also in various adult tissues such as bone marrow, adipose tissue, dental pulp, liver, and peripheral blood [1,2]. Treatment of exogenous MSCs is of increasing interest as a therapeutic tool for diseases, especially graft-versus-host diseases because MSCs are known to have a broad range of immune-modulatory and anti-inflammatory properties. However, transferring MSCs from bench to bedside is somewhat dampened due to the discrepancy between expected and actual results of MSCs as the therapeutic agent [3,4]. Recently, the investigation of the secretome of MSCs, instead of MSCs themselves, is burgeoning as an alternative therapeutic approach because it avoids the limitations of MSC-based therapy, including oncogenic transformation or immunoreactivity [5,6]. For instance, the secretome of MSCs has been proposed as a promising therapeutic approach for osteoarthritis (OA), which has long been considered untreatable. Notwithstanding its therapeutic potential, the exact underlying mechanism of how the secretome of MSCs exerts its therapeutic effects remains to be elucidated. Many researchers have made efforts to characterize the secretome of MSCs and to optimize culture conditions to fully exploit their therapeutic potential. To the best of our knowledge, studying the secretome of MSCs under co-culturing with other cells has been absent, even though it is well known that the biochemical composition of secretome is influenced by the environment to which the MSCs are exposed, as well as cell-to-cell communications. Investigation of secretome under co-culturing condition would provide more clues about the in vivo interaction between MSCs and their neighboring cells [7,8,9].

Proteomics studies about secretome are mostly conducted in vitro by culturing cells in serum-supplemented serum, such as 10% fetal bovine serum (FBS), and harvesting media to collect the secretome. However, with respect to the abundances of the secretory proteins, the excessive amounts of serum proteins hamper the proteomic analysis of the secretome with mass spectrometry [10,11]. In this regard, cells are transferred to the serum-free media before collecting the secretome. Serum deprivation, on the other hand, may disrupt cell metabolism and proliferation, affecting protein expression and secretion profiles [12]. To overcome this limitation, the bio-orthogonal non-canonical amino acid tagging (BONCAT) strategy can be adopted in secretome analysis. This technology supplements non-canonical amino acids, such as azidohomoalanine (AHA), to the cell culture media, allowing their incorporation into the newly synthesized proteins. AHA is the analogue of methionine bearing an azide, so the newly synthesized proteins can be captured by alkyne-conjugating resins based on click chemistry and an azide-alkyne cycloaddition reaction [12,13,14]. However, owing to the excess amount (5 mg/mL for 10% FBS), it is challenging to completely remove the serum proteins, which might non-specifically bind to polystyrene-plate or agarose resin.

AHA can be covalently linked to its cognate tRNA by the wild-type methionyl-tRNA synthetase (MetRS; AGA is not appropriate for labeling a specific cell in the co-culture system). In lieu of AHA, azidonorleucine (ANL) was selected. ANL is the methionine analogue having an azide group and requires the mutant MetRS (MetRS^L274G^) to be charged on tRNA. The charged tRNA can then be incorporated into the elongating polypeptide chains [15,16]. ANL-tagged proteins can be selectively conjugated and enriched through azide-alkyne cycloaddition. Therefore, if we express MetRS^L274G^ in a specific cell, then we can specifically enrich the newly synthesized proteins, which have ANL residues in their sequences, from that cel [17,18,19].

Here, we introduced the platform for investigating the secretory bodies of cells of interest, such as MSCs, co-cultured with other types of cells in serum-containing media. In this platform, the cells containing targeted secretome were transduced to express the mutant MetRS, driving any newly synthesized secretory proteins from the cells to achieve ANL residues in their sequences. Moreover, we introduced the approach to minimize possible false-positive identifications that are due to the high-abundance of serum proteins in the cell culture media. Last, with the developed platform, we characterized the secretome of MSCs, which were co-cultured with osteoarthritis (OA)-induced chondrosarcoma cells, and demonstrated their ability to inhibit OA.

## 2. Results

### 2.1. Cell-Type-Specific Labeling with Bioorthogonal and Non-Canonical Amino Acid

A strategy to analyze the secretome of specific cells that are co-cultured with other types of cells in serum-containing media is outlined in Figure 1A. For this cell-specific secretome study, we employed the BONCAT strategy and utilized ANL, which can be incorporated into a newly synthesized protein only in cells expressing mutant mouse methionyl-tRNA synthetase (MetRS^L274G^). The cells for which secretory proteins are analyzed are transduced to express MetRS^L274G^, while the other cells, co-cultured with the cells of interest, express the wild-type of MetRS that is incapable of using ANL during protein synthesis. Therefore, in the collected media after co-culturing two different types of cells, only proteins from the cells of interest contain ANL, which has an azide functional group and, therefore, can be enriched using alkyne-containing beads via click chemistry, a reaction between azide and alkyne. Proteins bound to the beads are then on-bead digested, indicating that the eluted peptides from the beads might have no ANL in their sequences. With this cell-specific BONCAT platform, we could distinguish the secreted proteins of the specific cells from any other contaminating proteins derived from either the cell culture media or the co-cultured cells.

We constructed various cell lines (chondrocyte cell line CHON-002 and placenta-derived MSC) expressing MetRS^L274G^ with a lentivirus transduction protocol (Figure 1B). The cells expressing the mutant MetRS were selected by treating them with puromycin (Appendix A). The expression of MetRS^L274G^ in the cells was confirmed by attaching the Myc tag to the N-terminus of the MetRS variant (Figure 1C). We further investigated the optimal concentration of ANL and the incubation time for efficient incorporation. The cell proliferation was affected upon treatments with different ANL concentrations from 0 mM to 2.5 mM (Appendix A). Compared to the experiment without ANL treatment (0 mM ANL), the number of cells was statistically significantly decreased in the treatment of more than 2 mM ANL (*p*-value < 0.05). We also investigated the ANL incorporation efficiency depending on the ANL concentration (1 mM or 1.5 mM) or the incubation time (12 h or 24 h). We lysed the transduced cells and tagged the lysate with the alkyne-activated biotinylation reagent. The incorporation efficiency, which was determined based on the biotin signal in the Western blot, increased as the concentration of ANL and the incubation time increased. (Appendix A). Considering both cell proliferation and incorporation efficiency, cells were incubated in the serum-containing media supplemented with 1.5 mM ANL for 24 h.

To assess whether the ANL-based enrichment method works properly, we examined the enrichment strategy using the lysate of CHON-002 cells expressing MetRS^L274G^ (CHON-002-MetRS^L274G^). In all analyses, except the application part in Section 2.4, we analyzed the samples by an LTQ-Orbitrap XL mass spectrometer with a short gradient time. In analyzing 1 µg of the cell lysate without enrichment, regardless of ANL treatment or no treatment, we identified approximately 600 human proteins (Figure 1D). However, after enrichment, in the case of the cells without ANL treatment, the identification count decreased to 12, while for the cells incubated with 1 mM ANL, we identified 513 human proteins. We further assessed the enrichment efficiency in the presence of background interference. A high amount of unlabeled proteins (10 mg of FBS-derived proteins) was introduced to the lysate of CHON-002-MetRS^L274G^. The interference resulted in lower protein identification counts, regardless of ANL treatment or no treatment (Figure 1D). The high-abundance serum proteins might mask the presence of low-abundance secreted proteins, making their detection and identification by a mass spectrometer difficult. However, despite the background noise, we identified 421 human proteins from the cells treated with ANL after enrichment, indicating that the enrichment strategy based on click chemistry worked properly.

### 2.2. The Secretome Analysis in Serum-Supplemented Media

To distinguish secreted proteins from proteins derived from cell culture media, we conducted two independent searches; in each search, we used a different database, either the Uniprot human database or the Uniprot bovine database, against which mass spectrometry (MS) data after enrichment were searched (Figure 2A). However, the larger the database, the more false-positive identifications we could have. Therefore, we aimed to minimize false-positive identifications by applying the following criteria: Only human proteins assigned with two or more unique peptides were counted; if the identified peptide matched to a human protein was also matched to a bovine protein, the following additional criteria were applied (Figure 2B): (a) For a human protein with two unique peptides, when one of the two peptides was matched to a bovine protein that was also identified with two unique peptides, we excluded the human protein from the final identification list, assuming that the shared peptide was derived from the bovine protein (Figure 2B, top); (b) If there was one shared peptide between the human and bovine proteins, but the bovine protein had no other unique peptide, the shared peptide was assumed to be from the human protein, indicating the presence of the human protein in the sample (Figure 2B, middle); (c) Last, when a human protein had one shared and two unique peptides, while the bovine protein matched by the shared peptide had another two unique peptides, then both proteins were assumed to be present in the sample, i.e., the human protein as the true-positive identification and the bovine protein as the contaminant (Figure 2B, bottom).

We examined the unspecific binding of proteins bearing no ANL in their sequences to the agarose resin. We processed 15 mL of fresh cell culture media containing 10% FBS through the alkyne-containing resin, thoroughly washed the resin, and performed the on-bead digestion. The MS data of the eluent was searched only against the Uniprot human database, and we identified 36 human proteins (Figure 2C). However, from the two independent searches against either human or bovine databases, having applied the criteria to exclude the shared peptides, we identified 66 bovine proteins (Figure 2D). Included among the identified bovine proteins were serum albumin, alpha-fetoprotein, alpha-2-HS-glycoprotein, or apolipoprotein A-I. Our group previously investigated the unspecific bindings of FBS proteins to the cell culture plate, and we listed the proteins that were unspecifically bound to the plate and composed the common repository of FBS proteins (cRFP) database [10]. Following the rationale of using the common repository of adventitious proteins in the analysis of the secretome from cells incubated in media with 10% FBS, the cRFP database could be utilized in the data search to eliminate the random matching of mass spectra that originated from the contaminant proteins to the secreted proteins from the cells of interest. Seventy percent of the bovine proteins we found in our study were found in the cRFP database.

We applied the aforementioned criteria to the analysis of the transduced cell line (CHON-002-MetRS^L274G^), which was cultured in media containing 10% FBS. We processed 30 µg of the cell lysate by means of the cell-specific BONCAT strategy. When the MS data were searched against only the Uniprot human database, we identified 687 human proteins (Figure 2C). In contrast, from the two separate searches with the Uniprot human or bovine databases, the number of identified human proteins decreased to 218, while the number of bovine proteins increased to 272 (Figure 2D). In the former setting, the spectra from the bovine serum proteins might be assigned to human proteins because of the shared peptides between human and bovine proteins. The analysis of the secretome of CHON-002-MetRS^L274G^ showed a similar trend (Figure 2C,D). Taken together, even though the BONCAT strategy was employed, we found that the high-abundance serum proteins of the cell culture media could not be thoroughly eliminated, thereby interfering with the secretome analysis.

### 2.3. Optimization and Validation of the Co-Culture System

Heretofore, we established the experimental conditions for the efficient incorporation of ANL and introduced the approach to minimize false-positive identifications. We further sought to acquire a deep secretome profile by adjusting the volume ratio between the media and the agarose beads. We loaded different volumes of the conditioned media to the alkyne-conjugated beads (slurry form in 2 mL) and performed the enrichment procedure. With 20 mL of the conditioned media, we identified the highest number of the secretory proteins (76 human proteins), which represented the highly true-positive identifications due to the implementation of the criteria described in Section 2.2 (Figure 3A). With the optimal volume, we conducted the enrichment procedure in technical triplicate. As a result, 70.3% of the peptides and 79.2% of the proteins were identified in at least two out of three replicates, showing the reproducibility of our strategy (Figure 3B). Furthermore, the quality of our secretome profile was assessed using SignalP, SecretomeP, and TMHMM, which predict the protein localization in silico based on specific sequence motifs. Of the identified proteins in the secretome analysis, 86.6% of the proteins were predicted as being secretory (Figure 3C). Moreover, for the identified proteins, knowledge-based Gene Ontology Cellular Compartment (GOCC) enrichment analysis was performed, and among the significantly enriched GOCC terms, the top 10 ranked terms were all related to extracellular localization (Figure 3D). Collectively, our platform is reproducible and provides reliable identification of cell-specific secretome, without any experimentally biased identifications that were possibly derived from cell culture media or cytoplasmic proteins from cells.

Using the BONCAT strategy and the established database search method, we could discriminate between the secreted proteins and the serum proteins in the cell culture media. However, any proteins that stemmed from the co-cultured cells, due to cytolysis or secretion from the cells, might be mis-annotated as secreted proteins from cells of interest. As a proof-of-concept, we designed a two-proteome system by co-culturing two different cell types from different organisms: human CHON-002-MetRS^L274G^ and mouse ATDC5 cells. The MS raw file acquired after enrichment was searched against Uniprot human, mouse, or bovine databases in the separate mode, as before (Figure 4A). We identified a total of 990, 397, or 1246 peptides from the searches, utilizing Uniprot human, mouse, or bovine databases, respectively (Appendix A). Then, we applied the criteria mentioned in Section 2.2 to exclude any peptides matched to bovine proteins from each result, resulting in 633 human and 167 mouse peptides (Appendix A). We compared the sequences between the human peptides and the mouse peptides and found that 124 peptides were shared between the two organisms (Appendix A). We excluded these 124 peptides in further analysis because we could not determine whether their origins were human or mouse proteins. Based on the filtered peptides that were assessed as unique to each organism with high confidence, we identified 80 human proteins and six mouse proteins (Figure 4B), noting that the interference by the co-culturing cells in our strategy was minimal. 

In order to acquire the deeper secretome profile, we employed a fractionation strategy based on high pH reversed-phase chromatography. After the enrichment of 20 mL of the conditioned media and the on-bead digestion, we fractionated the peptides into 12 fractions. With this fractionation, we identified more than four times the human proteins (363 proteins), compared to the number of proteins from the single-shot analysis (80 proteins) (Figure 4B). With fractionation, we still acquired the low number of identified mouse proteins (Figure 4B). Moreover, the quality of the secretome data acquired after the fractionation was comparable to that of the single-shot analysis, with 72.7% of the identified proteins being secretory proteins (Figure 4C). Taken together, with the optimal volume of the conditioned media and the fractionation strategy, we could identify 363 secreted proteins with high reliability.

### 2.4. Application of the Co-Culture System; Analysis of MSC Secretome

To assess whether we could characterize the therapeutic effect of the secretome of MSCs using our strategy, we co-cultured chondrosarcoma cells (SW1353) that were treated with a pro-inflammatory cytokine, interleukin 1 beta, to induce OA, with the MSCs expressing MetRS^L274G^ (Figure 5A), and we profiled the secretome of the MSCs analyzing the secretome in terms of its ability to suppress OA (Appendix A). The MSCs were isolated from placenta (PL-MSCs), and we confirmed the expression of the mutant MetRS in the MSCs by replacing the gene of the mutant MetRS with the gene of green fluorescent protein (Appendix A). Based on the literature search, we listed 30 genes for which the expression levels were reported to alter as OA progressed (Table 1). When OA was processed, some of the genes, such as MMP13 or IL6, were reported to increase, while others [20,21,22], such as COL2A1, decreased [23]. Among the genes related to OA, we measured the expression levels of 14 genes in the co-cultured SW1353 cells, based on the quantitative polymerase chain reaction (qPCR) (unpublished data). We quantified the gene expression levels in a total of 26 co-culturing combinations (MSCs and SW1353). Some of the SW1353 cells expressed genes in the opposite direction as that of the OA-induced cells, showing that the MSCs suppressed OA in these SW1353 cells. The top three MSCs that effectively inhibited OA were designated as “good MSCs”, while the top three MSCs with poor OA-inhibiting power were designated as “poor MSCs”. 

In order to evaluate whether the secretome analysis can predict the therapeutic effect of MSCs, we analyzed the secretome from the six MSCs (three from good MSCs and three from poor MSCs) with our enrichment strategy, using Q-Exactive. After enrichment, we identified average 586 and 574 proteins in the secretome from the good and the poor MSCs, respectively, with an intergroup overlapping percentage of 85% (Figure 5B). We measured the abundances of the identified proteins in the secretome using a label-free quantification (LFQ) approach and compared the expression levels of the proteins between the two groups (good vs. poor). We found that the abundances of 26 proteins between the two groups were statistically significantly different (1.5-fold-change, *p*-value < 0.05) (Figure 5C). With the secretome data, we predicted any perturbation in the expression levels of genes related to OA using Ingenuity Pathway Analysis (IPA) software. We input the genes of 26 differentially secreted proteins to the IPA tool. For 12 of the 26 genes, their downstream pathways were related to OA (Figure 5D). We acquired multiple genes predicted to be either upregulated or downregulated by the secretome of good MSCs compared to poor MSCs (Figure 5D). For instance, the expression levels of COL2A1 or ACAN were predicted to increase when influenced by good MSCs compared to poor MSCs.

The prediction result was quite comparable to the qPCR data. Of the genes whose expression levels were measured by qPCR, five genes were estimated to show different expressions between good MSCs and poor MSCs. Among the five genes, the predicted expression levels of four genes (MMP3, MMP13, BAX, and TNFα) were correlated with the qPCR results. In contrast to the qPCR result, the remaining gene, TGFβ, was anticipated to be lower in good MSCs compared to poor MSCs, but in other studies, it was reported that the expression level of TGFβ was lower in normal cells than in OA cells, which had a correlation with our prediction data [24]. Therefore, we further compared our prediction with the literature (Table 1). Among the 30 genes related to OA, 21 genes showed different expressions in good MSCs compared to poor MSCs, and the expected alterations of 15 genes between good MSCs and poor MSCs were positively correlated with the reported alterations of the expression levels in normal chondrocyte cells and OA-induced cells in the literature (normal chondrocyte cells vs. OA-induced cell). These comparisons (predicted alterations by secretome vs. qPCR data, or vs. literature) indicated that our secretome data accurately described MSCs’ ability to inhibit OA.

**Table 1 ijms-23-06527-t001:** Genes reported to alter depending on osteoarthritis (OA) pathology.

Gene Name	Normal Chondrocyte Cell (vs. OA-Induced Cell) ^¶^	Predicted Alterations by Secretome vs. Literature ^$^	Reference
MMP3	−	Mat.	[25,26]
MMP13	−	Mat.	[20]
ADAMTS5	−	Opp.	[25]
Caspase9	−	Mat.	[27]
Bax	−	Mat.	[28,29]
Bcl-2	N.C	Mat.	[30]
COL2a1	+	Mat.	[23,25]
ACAN	+	Mat.	[23]
SOX9	+	N.D	[23]
HIF1α	+	Opp.	[31]
TNFα	−	Mat.	[32]
IL-1β	−	N.D	[33]
IL-4	+	Mat.	[34]
TGFβ	−	Mat.	[24]
COL1a1	−	Opp.	[23]
BMP2	−	N.D	[23]
RUNX2	−	Opp.	[23,35]
COL10a1	−	N.D	[23,36]
IHH	−	Mat.	[23,37]
DKK1	+	N.D	[23,38,39]
FRZB	+	N.D	[23]
GREM1	+(early)/−(late)	N.D	[23]
AXIN2	−	N.D	[23]
TIMP1	+	Mat.	
IL-6	−	Mat.	[21,22,40]
LIF	−	N.D	
IL-17	−	Mat.	[33]
MMP1	−	Mat.	[20,25]
NF-kB	−	Opp.	[22,41]
STAT3	−	Opp.	[21,42]

¶, The expression levels of genes in normal chondrocyte cells compared to osteoarthritis-differentiated cells reported in the literature. + indicates that the gene level was higher in normal cells than in OA-differentiated cells. − indicates the opposite case. $, Comparison between the expression levels in normal chondrocytes reported by other studies and the predicted levels based on our secretome profile using IPA tool; N.C., no change; Mat., IPA-predictions matched to literature; Opp., IPA-prediction contradictory the literature; N.D., no decision.

## 3. Discussion

In this study, we presented a cell-type-specific tagging technique, enabling the investigation of the secretome from specific cells that were co-cultured with another type of a cell line in serum-supplemented media. This technique employed one of the methionine analogues, azidonorleucine (ANL). ANL bears an azide functional group that can be ligated to an alkyne through Cu(I)-catalyzed azide-alkyne cycloaddition. In order for cells to utilize ANL in protein synthesis, they should harbor a mutant methionyl-tRNA synthetase (MetRS). With the first-introduced variant of Escherichia coli MetRS, which is capable of charging ANL to tRNAs, ANL was selectively incorporated into protein N-terminal sites, not internal sites in mammalian cells [43,44]. In this regard, we employed the mouse MetRS variant (MetRS^L274G^) to label mammalian cell proteins with ANL. According to the research team that introduced the mouse mutant MetRS, MetRS^L274G^ had greatly enhanced activity toward ANL and improved discrimination against Met [16]. By expressing the mouse MetRS^L274G^ in cells for which the secretome is of interest, we enriched the secreted proteins of the cells based on affinity chromatography. With the transduced mammalian cells, we noticed that the treatment of highly concentrated ANL adversely affected cell proliferation while lowering the ANL concentration, resulting in inefficient ANL incorporation. In this study, we determined the optimal ANL treatment condition during the co-culturing experiment (1.5 mM ANL for 24 h) without altering cell proliferation, achieving efficient ANL incorporation.

In the analysis of secretome from specific cells in the co-culture system, there are two sources, the cell culture media or the other co-cultured cells, from which proteins might distort the results obtained for the secretome analysis. First, when it comes to serum proteins in the media, it was reported that the amounts of the serum proteins in the 10% FBS-supplemented media were 1000 or 10,000 times greater than the amounts of the secreted proteins [45]. Regarding the high-abundance serum proteins, researchers transferred the cells from the serum-containing media until they acquired enough cells, then transferred the cells to the serum-free media just before collecting the secretome. Even though cells tolerate starvation conditions for a short period of time (12–48 h), this condition might affect cell growth, distorting protein expression and the secretion profile. Our group previously reported that even within the same cell type, secretome patterns varied according to culture conditions [12]. Therefore, it was necessary to harvest the secretome in the presence of the background serum-contaminating proteins.

The BONCAT strategy would relieve the interference issue somewhat, because proteins derived from the media theoretically have no ANL in their sequences, so they cannot be captured by the alkyne-containing agarose resin. However, it is possible that the proteins without ANL unspecifically bind to the agarose resin, remain on the bead even after thoroughly washing the resin, and are co-eluted with the target peptides. For the serum proteins, due to their high abundance, their unspecific binding to the resin might be problematic. Therefore, to reduce false-positive identifications due to this experimental bias, we established the criteria for excluding any identified peptide that was also matched to bovine proteins. After applying the criteria at the peptide level, we considered the identified human proteins as true-positive identifications only if they had two or more unique peptides. Due to the quite stringent criteria, the number of identified human proteins decreased. However, the filtered list would contain more reliable identifications.

We then evaluated the interference by the cells, co-cultured with the cells of interest. For distinguishing among proteins originated from different cell types, we expressed the mouse mutant MetRS only in one cell type. However, the enriched peptides, on-bead digested and eluted from the alkyne-containing column, would have no ANL in their sequences, indicating that there was no direct evidence that the eluted peptides initially originated from the target proteins bearing ANL residues. Even when strict precautions were followed, handling cells could burst the cells and release the intracellular proteins, which might unspecifically bind to the alkyne-containing column, leading to bias. In this regard, in order to evaluate this possible disturbance, we designed the two-proteome model, in which two distinct cells from different organisms were co-cultured in the serum-containing media. We employed the mouse cells which expressed only wild-type MetRS that was unable to use ANL during protein synthesis. In the analysis of the secretome of human cells that were co-cultured with the xenogeneic cells, we confirmed that the interference by the cells, which were co-cultured with the cells of interest, was minimal. Among the peptides that were regarded as bovine or mouse peptides, and thus excluded from the final list, some of the peptides might actually be derived from human proteins. However, we performed further analysis using only peptides that were human, with high confidence, to obtain a reliable result. Even after filtering out all the identified peptides that were regarded with low confidence, the depth of our secretome profile, especially when it was acquired after high-pH reversed-phase fractionation, was comparable to other studies regarding the secretome [46].

This co-culturing system is accommodating for profiling the secretome of MSCs, which are co-cultured with disease-related cells in serum-containing media. The co-culturing condition would more closely reflect the environment in vivo and enable the evaluation of the possible crosstalk between MSCs and their neighboring cells. Moreover, it has been reported that the secretome composition of MSCs can be influenced by the environment to which the MSCs are exposed. Osteoarthritis (OA) is now considered curable by the treatment of MSCs [47,48,49,50]. Along with this cell-based therapy, cell-free therapy using the secretome of MSCs is regarded as a promising therapeutic approach for OA [51]. Many researchers have co-cultured MSCs and chondrocyte cells in order to study the secretome of MSCs and their effect on OA [38,52,53]. They discovered that upon co-culturing with MSCs, the proliferation of chondrocyte cells increased, while the apoptosis decreased. Moreover, it has been reported that the therapeutic effect for OA differs, depending on the source of MSCs [20,54,55,56,57]. Cord blood-derived or Wharton’s jelly-derived MSCs (WJ-MSCs) were more effective for treatment than bone marrow-derived MSCs [58]. When chondrocytes or synoviocytes were co-cultured with MSCs, the expression of several genes known to induce OA-pathology decreased. These genes included MMP-1, MMP-3, MMP-13, and IL-6. In this co-culturing condition, the expression levels of aggrecan, sox-9, collagen type II, and TGF-1β decreased. However, in this co-culturing condition, it was difficult to analyze only secreted proteins from MSCs, complicating the determination of which proteins of MSCs were delivered to the target cells and regulated the signaling pathway.

In this study, using our platform, we investigated the effect of the secretome derived from human placenta-MSCs (PL-MSCs), which were co-cultured with OA-induced cells. By analyzing the secretome of in vitro cultured MSC populations, we were able to understand the mechanism of how the secretome of MSCs exerts its therapeutic effects. We examined a total of 27 co-culturing sets (PL-MSC expressing MetRS^L274G^ and OA-induced chondrosarcoma cell). Although all MSCs showed inhibiting ability on OA progression, we selected three MSCs (good MSCs) that showed the most OA-inhibiting ability compared to poor MSCs that had the lowest inhibiting ability. We compared the secretome profiles between the good and poor MSC groups. Of the identified proteins, the abundance of IL11 was statistically significantly higher (log2 fold change of 2.1) in good MSCs than in poor MSCs (Figure 5C). According to IPA, the high expression of IL11 would induce the activation of COL2A1, the gene encoding collagen type II which is the major component of the cartilage matrix and is known to be degraded as OA proceeds. In addition, the elevated IL11 would deactivate pro-inflammatory cytokine (IL6) and metalloproteases (MMP3/MMP11). In other studies, the degradation of cartilage was accelerated by the over-activation of matrix-degrading enzymes, which was induced by the activated pro-inflammatory pathways. This indicated that IL11 is a possible therapeutic agent for OA. Collectively, the prediction of OA-related gene perturbation by MSCs based on our secretome profile was highly correlated to the observed alteration when OA was inhibited. We expect that our platform can be utilized to determine which MSCs would have better therapeutic abilities, based on the analysis of the secretome of MSCs (Figure 6).

## 4. Conclusions

We presented the strategy to profile the cell-specific secretome under culturing of two different cell types by enabling only the specific cells to use the non-canonical amino acid ANL during protein synthesis. In our strategy, it was not necessary to transfer the cells from the serum-containing media to serum-free media before harvesting the secretome to avoid any experimental bias caused by the highly abundant serum proteins, so it could provide the unbiased landscape of secretome. In addition, we confirmed that interference by other cultured cells, which were unable to use ANL, was minimal, and this was consistent with the assumption that the enriched proteins really did contain ANL residues in their sequences. With our developed platform, in the analysis of the secretome derived from MSCs, which were co-cultured with OA-induced cells, we showed that secretome analysis could predict the therapeutic effect of the MSCs and determine which MSCs would have better therapeutic ability. Moreover, considering that the secretome of MSCs appears to vary significantly, depending on the niches where the MSCs reside, our secretome analysis platform will be useful for us to gain a better understanding of the therapeutic effects of MSCs in vivo, so as to provide a better treatment option for patients.

## 5. Materials and Methods

### 5.1. Cell Culture

CHON-002, SW1353, ATDC5 cell lines were purchased from ATCC, and placenta-derived MSCs (PL-MSCs) were provided by Prof. Chul-Won Ha. All cell lines were cultured in DMEM-low glucose (Gibco, Waltham, MA, USA) medium with 10% fetal bovine serum (FBS) and incubated at 37 °C in a humidified atmosphere containing 5% CO_2_. The medium was replaced every 2–3 days in a tissue culture plate. CHON-002 and PL-MSCs were transduced to express the mouse MetRS mutant. In the case of PL-MSCs, they were grown to passage 7 and then used for the co-culturing experiment.

### 5.2. Construction of a Cell Line Expressing Mutant Methionyl-tRNA Synthetase (MetRS)

A plasmid (pMarsL274G) that encodes the mutant mouse MetRS gene was purchased from Addgene and was ligated to the pLenti-EF1a-CMyc-DDK-IRES-Puro (Origene™) vector. The vector was PCR-amplified using an Asc Ⅰ forward primer and a Mlu Ⅰ reverse primer and was transformed into E. coli MegaX DH10B (Qiagen, Redwood city, CA, USA) competent cells. The cells were spread in LB agar plate, and the colonies expressing the mutant MetRS were selected using 100 µg/mL ampicillin. Plasmid DNA was purified using a Mini-prep kit (Qiagen, Venlo, The Netherlands) and further confirmed by sequencing (Laragen, Virginia Ave, CA, USA). For lentivirus infection, CHON-002 and PL-MSCs were seeded to approximately 60% confluence and treated with lentivirus by adding polybrene (8 μg/mL). The infected cells were incubated for 48 h at 37 °C in a humidified atmosphere containing 5% CO_2_, and then the media were replaced with fresh media. MetRS^L274G^ expressing cells were selected using 2 µg/mL of puromycin.

The expression of the mutant MetRS in the transduced cells was confirmed using western blot analysis. In this case, myc tag was co-expressed at the N-terminus of the mutant MetRS. The transduced cells at 70–80% confluency were washed three times with cold phosphate-buffered saline (PBS) and then incubated in cell culture medium containing ANL (0 to 2.5 mM) for 12–24 h. After that, the medium was removed and washed three times with PBS. A lysis buffer (8M urea, 50 mM Tris, pH 8.0, and 75 mM NaCl) with EDTA-free protease inhibitor (Roche, Basel, Switzerland, 1187358001) was added, and the cells were detached from the cell plate using a scraper. The cells were transferred to a tube and sonicated on ice for 30 s using a probe sonicator. The sample was centrifuged at 15,000× *g* for 10 min at 4 °C, and the supernatant was transferred to a new tube. Protein concentration was measured with bicinchoninic acid (BCA) protein assay (Pierce™, Waltham, MA, USA).

The lysate was then separated by 0.1% sodium dodecyl sulfate polyacrylamide gel electrophoresis. The separated protein bands were transferred to a nitrocellulose membrane. To prevent non-specific binding, the membrane was blocked overnight using 5% nonfat milk in Tris-buffered saline containing 0.05% Tween-20 (TBST). The membrane was then incubated with mouse anti-Myc (1:1000; Cell Signaling Technology, Danvers, MA, USA) primary antibody for 1 h at room temperature. After washing the membrane three times with TBST, the blots were incubated with the appropriate horseradish peroxidase-conjugated (HRP-linked) secondary antibody. After three additional washes with TBST, the membrane was developed using an enhanced chemiluminescent system (Amersham, Buckinghamshire, South East, England). The incorporation of ANL into the newly synthesized proteins was also confirmed by western blot analysis. The protein extracts of the transduced cells treated with ANL were subjected to click reaction according to the manufacturer’s manual of the Click-iT Proteins Reaction Buffer kit (Invitrogen, Waltham, MA, USA). The following procedures were as described above, but in this case, anti-biotin HRP-linked antibody (1:2000; cell signaling technology) was used.

### 5.3. Sample Preparation for the Click Chemistry-Based Enrichment: Cell Lysate and Secretome

In order to label any newly synthesized protein from the transduced cells with ANL, the transduced cells at 70–80% confluency were washed with cold PBS and incubated in DMEM-low glucose media containing 10% FBS and 1.5 mM ANL for 24 h. With these transduced cells, we prepared two types of ANL-tagged protein samples, cell lysate or secretome, for the click chemistry-based enrichment. In the case of preparing the cell lysate, after washing the cells twice with PBS, they were collected using a scraper in PBS, containing protease inhibitors without EDTA, into a tube. To obtain the cellular proteins, the cells were sonicated three times with a probe sonicator on ice for 20 s. The protein extract was alkylated with 10 mM iodoacetamide (Pierce) for 30 min in the dark at room temperature to block thiol-ene reaction. For the cell lysate used to examine the enrichment efficiency in the presence of background noise, thiol-blocking reaction was conducted after FBS-derived proteins were introduced to the cell lysate. 

To collect the secretome of the transduced cells, 20 mL of conditioned media, in which the transduced cells were incubated, was transferred to a tube, and EDTA-free protease inhibitor was added to the media, followed by centrifugation at 1000 g for 30 min at 4 °C. Then, the media, without any cell debris, was carefully passed through a 0.22 µm filter and 9.01 g of urea was added to the filtrate to achieve a final urea concentration of 6 M. In order to block the thiol group before the click reaction, 200 mM chloroacetamide (using 1 mg/mL chloroacetamide dissolved in PBS) was treated, and the sample was incubated in the dark for 60 min at room temperature. The sample was then concentrated to 500 µL, using a 10 kDa MWCO centrifugal filter (Merck, Darmstadt, Germany, UFC900396).

### 5.4. Secretome Preparation in Co-Culture Condition

On the first day, SW1353 cells were seeded in a 6-well plate (5 × 10^5^ cells/well) and cultured for 24 h. On the second day, the medium of SW1353 cells was replaced with 2.5 mL of fresh medium containing 10 ng/mL of IL-1β and SW1353 cells were cultured for 24 h. In another 6-well plate filled with 3 mL of medium, the trans-well was inserted. The transformed MSCs were seeded inside the trans-well (2 × 10^5^ cells/trans-well) and incubated for 24 h. On the third day, 1.5 mM ANL was added to SW1353 cells. The trans-well medium, in which MSCs were cultured, was replaced with 1.5 mL of fresh medium supplemented with 10 ng/mL IL-1β and 1.5 mM ANL, followed by inserting the trans-well into a plate in which SW1353 cells were cultured. After co-culturing the two different cells for 24 h, all media from the plate and the trans-well were transferred to a tube. The secretome from this media was harvested following the procedures described above. In the co-culture experiment using xenogeneic cells, ATDC5 and CHON-002-MetRS^L274G^ without treatment of IL-1β were used instead of SW1353 and MSC-MetRS^L274G^, respectively.

### 5.5. Enrichment of ANL-Tagged Proteins Using Click Chemistry

The click chemistry reaction was performed with the Click-iT protein enrichment kit (Invitrogen). The sample was mixed with 200 µL of the bead slurry, which had been washed three times with water. Then, the volume of the sample was adjusted to 1 mL using the urea lysis buffer of the kit. The sample was mixed with 1 mL of 2× catalyst solution and incubated at room temperature for 18 h. The sample was centrifuged at 1000× *g* for 2 min at room temperature, and the supernatant was carefully removed. The beads were washed three times with 1 mL of water. 1 mL of 1% SDS containing 20 mM DTT was introduced to the sample, and the sample was incubated at 70 °C for 20 min and cooled at room temperature for 15 min. After centrifugation, the supernatant was removed, and the beads were washed with 1 mL of water. The alkylation of the proteins bound to the beads was performed by incubating it with 1 mL of 1% SDS containing 40 mM iodoacetamide for 30 min in the dark. The beads were thoroughly transferred to the column provided by the kit. The beads in the column were washed three times with 2 mL of water and then ten times with 2 mL of 1% SDS. In addition, washing was repeated 15 times with 2 mL of each of 8M urea/100 mM Tris, 20% isopropanol, and 20% acetonitrile, in order. All washing processes were performed with the gravity flow. The column was capped, and the beads were re-suspended in 500 µL of digestion buffer (100 mM Tris (pH 8.0), 2 mM CaCl2 and 10% acetonitrile). The beads were moved to a 1.5 mL tube, and another 500 µL of digestion buffer was used to transfer any left beads in the column to the tube. After centrifugation at 1000× *g* for 2 min at room temperature, 800 µL of the supernatant was removed. 10 µL of trypsin (0.1 µg/µL) was added to the tube, and it was incubated at room temperature for 16 h. The beads were transferred to a 0.22 µm centrifugal filter and centrifuged to collect the peptide; 500 µL of water was additionally added to the filter, and the filtrate after centrifugation was merged into the tube containing the digest, and this was repeated one more time. The filtrate was acidified using 20 µL of 10% TFA for desalting with a C18 cartridge.

### 5.6. Basic pH Reversed-Phase Fractionation

Peptides were fractionated by basic pH reverse phase liquid chromatography (bRPLC) using an Agilent 1290 Infinity LC system (Agilent Technologies, Santa Clara, CA, USA). The fractionation was performed on an XBridge BEH130 C18 column (250 mm × 4.6 mm, 3.5 µm, 130 Å; Waters, Milford, MA, USA) with a flow rate of 0.5 mL/min. Mobile phases were 10 mM ammonium formate (pH 10) as buffer A and 10 mM ammonium formate in 90% acetonitrile (pH 10) as buffer B. The gradient was 2–5% buffer B for 10 min, 5–40% buffer B for 40 min, 40–70% buffer B for 15 min, 70% buffer B for 10 min, and 70–5% buffer B for 15 min. Fractions were collected every 0.8 min, resulting in 120 fractions in total. The fractions were then concatenated into 12 fractions by combining every twelfth fraction together (tube 1 consisted of fractions 1, 13, 25, 37, 49, 61, 73, 85, 97, and 109). The merged 12 fractions were than dried using speedVac.

### 5.7. Liquid Chromatography—Tandem Mass Spectrometry (LC-MS/MS) Analysis

In this study, we employed two different mass spectrometers, either LTQ Orbitrap XL (Thermo Fisher Scientific, Waltham, MA, USA) or Q-Exactive (Thermo Fisher Scientific). All of the analyses, except the analysis of MSC secretome, were performed with LTQ Orbitrap XL. In the case of LTQ Orbitrap XL coupled with the Eksigent NanoLC-2D system, the dried peptide samples were reconstituted in 0.1% formic acid and separated on a microcapillary column (15 cm × 75 µm I.D.) which was packed with Reprosil-Pur C18-AQ 5 μm resin (Dr. Maisch GmbH, Ammerbuch-Entringen, Germany) column. The mobile phases A and B were 0.1% formic acid in water and 0.1% formic acid in acetonitrile, respectively. The sample was eluted with 5% solvent B for 15 min, 5–40% solvent B for 210 min, and 30–50% solvent B for 20 min at a flow rate of 300 nL/min. Total run time was 275 min. Full scan MS spectra (300–1500 m/z) were acquired with a resolution of 6 × 10^5^, automatic gain control of 1 × 10^5^, and a maximum ion time of 100 ms. In each full MS scan, up to the seven strongest ions were fragmented and analyzed. The MS/MS parameters were set as follows: an automatic gain control target of 3 × 10^5^, a maximum ion time of 50 ms, an isolation window of 2.0 m/z, normalized collision energy of 35, a dynamic exclusion period of 30 s, and an intensity threshold of 500.

In the analysis with Q-Exactive coupled with the Eksigent nanoLC-ultra 1D plus system, the dried peptide samples were reconstituted in 0.1% formic acid and separated on an EASY-Spray column (25 cm × 75 µm) at a flow rate of 300 nL/min. The mobile phases were the same in the setting of LTQ Orbitrap XL. The peptide was eluted using a linear gradient with 2% solvent B over 10 min, followed by a linear gradient from 2% to 30% solvent B over 110 min and from 30% to 70% solvent B over 3 min. The total run took 158 min. Full scan MS spectra (400–1800 m/z) were acquired with a resolution of 70,000, automatic gain control of 1 × 10^6^, and maximum injection time of 30 ms. In each full MS scan, up to 12 strongest ions were fragmented and analyzed. MS/MS parameters were set as follows: resolution of 17,500, automatic gain control of 5 × 10^4^, a maximum injection time of 120 ms, an isolation window of 2.0 m/z, normalized collision energy of 27, a dynamic exclusion period of 30 s, and an intensity threshold of 4.2 × 10^3^.

### 5.8. MS Data Analysis

The mass spectra were analyzed with the SEQUEST HT module in Proteome Discoverer 2.16 (Thermo Fisher Scientific). Data obtained by LTQ Orbitrap XL were analyzed with a precursor mass tolerance of 2 Da and a fragment mass tolerance of 0.6 Da. The data from Q-Exactive were searched with a precursor mass tolerance of 10 ppm and a fragment mass tolerance of 0.02 Da. The databases used in this study were the Uniprot human database (released in February 2020, http://www.uniprot.org), the bovine database (February 2020), and the mouse database (February 2020). Search parameter settings included full tryptic specificity with up to two missed cleavages, carbamidomethylation of cysteine as a fixed modification, oxidation of methionine, and acetylation of protein N-terminus as variable modifications. The false discovery rate (FDR) was set to 0.01 at both the protein and PSM levels. In order to acquire the abundance of peptides, a label-free quantification method was employed using a combination of the Minora Feature Detector with default settings, Feature Mapper, and Precursor Ions Quantifier modules in Proteome Discoverer version 2.16. In the Feature Mapper module, only peptides with a maximal retention shift below 5 min were accepted for quantification. Precursor abundance was measured based on peak area, and for each raw file, the total peptide amount was used to normalize the quantification result.

### 5.9. Bioinformatics Analysis

GO terms were analyzed using the algorithm of the DAVID tool (https://david.ncifcrf.gov/, accessed on 8 August 2021)). The cutoff parameter of the FDR for the analyzed GO terms was less than 0.01. Secretome analysis classified pathways using SignalP version 4.1, SecretomeP version 2.0, and THMHH version 2.0 provided by DTU bioinformatics. Proteins not included in the three prediction results were classified as “others”. Biological function prediction and protein network analysis were performed using the IPA program (Qiagen). A t-test was performed using the LFQ intensity, and proteins with a p-value less than 0.05 were used for IPA analysis.

## Figures and Tables

**Figure 1 ijms-23-06527-f001:**
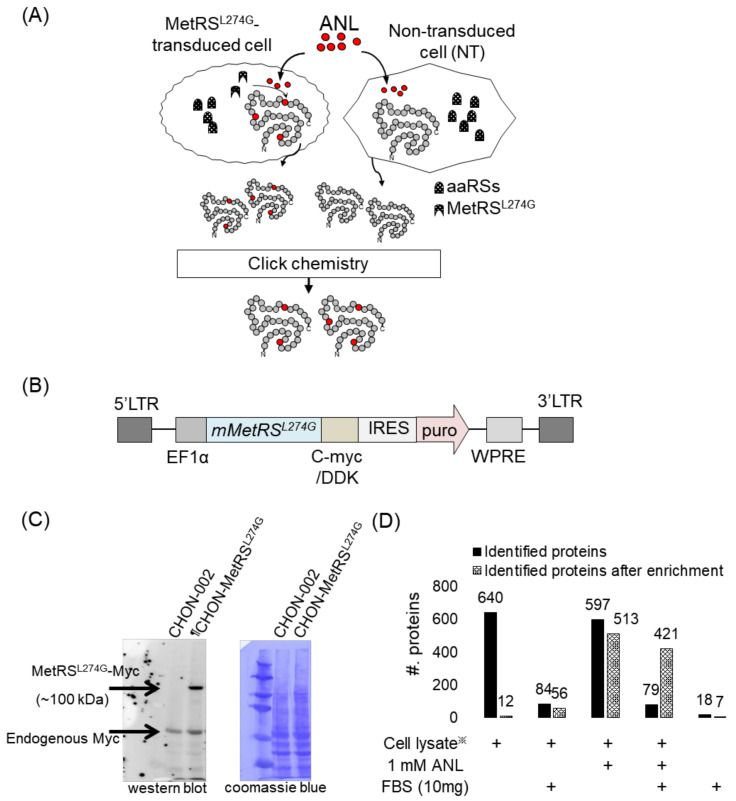
Labeling of a newly synthesized protein with ANL using a mutant mouse MetRS (MetRS^L274G^)-expressing cell line. (**A**) Our strategy is to label a newly synthesized protein in a specific cell that is co-cultured with another type of cell. The specific cell whose secretome is of interest is transduced to express a mouse MetRS variant (MetRS^L274G^) capable of ANL incorporation during protein synthesis. (**B**) Structure of the lentiviral vector used to express the mutant mouse MetRS^L274G^ gene. LTR: ; long terminal repeat; EF1α: elongation factor 1 alpha; mMetRS^L274G^: mouse mutant MetRS gene; DDK: FLAG tag; IRES: internal ribosome entry site; puro: puromycin resistance gene; WPRE: WHP posttranscriptional regulatory element. (**C**) Expression of MetRS^L274G^ in the transduced cell. The expression of MetRS^L274G^ is confirmed using an anti-Myc antibody in a western blot with mutant MetRS tagged with Myc at its N-terminus (left). The protein expression pattern in the transduced cell line does not differ from the one in wild-type cells according to Commassie blue staining (right). (**D**) Verification of our enrichment strategy. The lysate of MetRS^L274G^-expressing cell, which is either treated with ANL or not treated, is processed by click chemistry-based enrichment, and the number of identified proteins is indicated. FBS-derived proteins are used as background interference. ※: MetRS^L274G^ transduced CHON-002 cell lysate.

**Figure 2 ijms-23-06527-f002:**
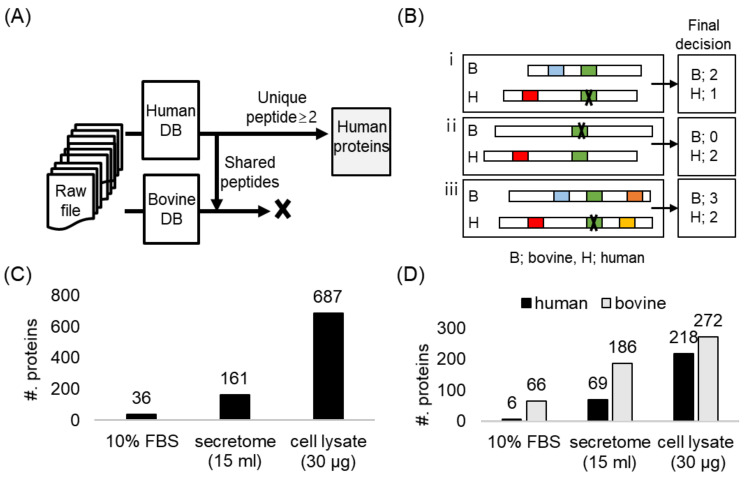
Analysis of secretome in 10% FBS-supplemented media. (**A**) A schematic diagram of two independent searches using human and bovine databases to minimize the false-positive identification possibly derived from 10% FBS. (**B**) Criteria to determine the origin of shared peptide between human and bovine proteins. Each horizontal bar indicates the whole sequence of a protein. The color box inside the bar indicates the peptides identified in each database, and the green box indicates the shared peptide between human and bovine proteins. Cross marks in black indicate that the peptide is assumed not to be derived from this protein. In the final decision, only proteins with two or more peptides are counted. (i) it was determined that only bovine protein was identified, (ii) determined that only human protein was identified, (iii) both bovine and human proteins were determined to be identified. (**C**) Number of identified proteins after enrichment using only the human database. (**D**) Number of identified proteins after enrichment using both the human and bovine databases and applying the criteria.

**Figure 3 ijms-23-06527-f003:**
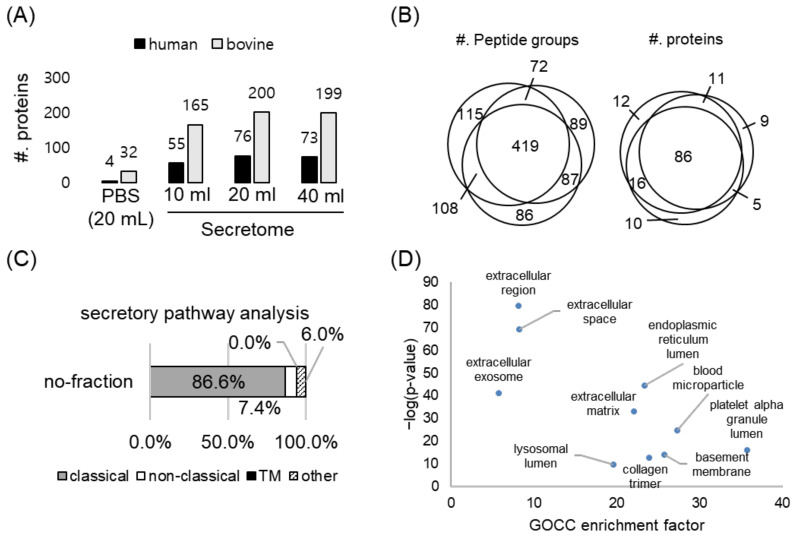
Enrichment of ANL-labeled proteins in the secretome. (**A**) Optimization of the ratio between the conditioned media volume and the alkyne-containing bead volume. In our strategy, 15 mL of the conditioned medium is the optimal point in terms of the depth of the secretome profile. (**B**) Reproducibility of our enrichment strategy: 79.2% (or 70.3%) of proteins (or peptides) were identified in at least two out of three replicates. (**C**) The secretory pathway analysis of the identified proteins in secretome after enrichment using SignalP, SecretomeP, and TMHMM. (**D**) Gene ontology cellular component analysis of the identified proteins in secretome after enrichment.

**Figure 4 ijms-23-06527-f004:**
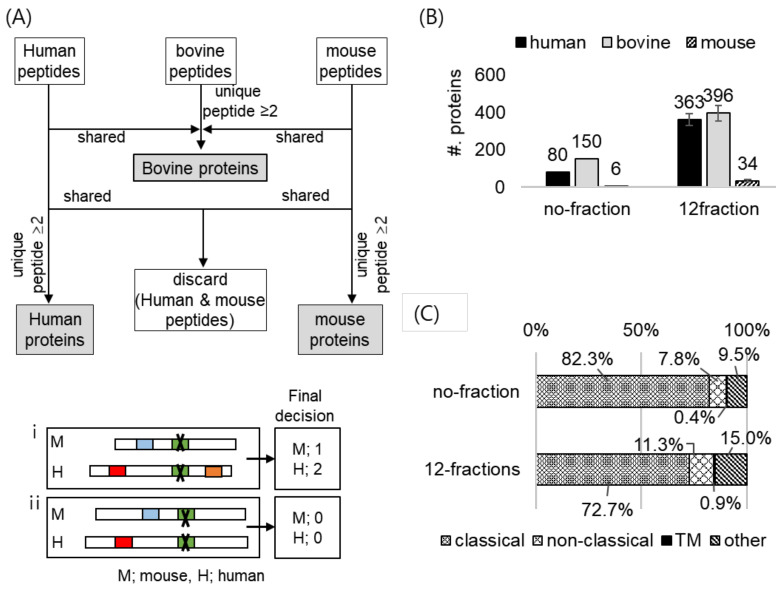
Investigation of any interference by cells that are co-cultured with our target cells. The transduced human CHON-002 cells were co-cultured with the xenogeneic cells, which are mouse ATDC5 cells that cannot use ANL during protein synthesis. (**A**) Schematic diagram of decision on the origin (bovine, mouse, or human) of the shared peptide (top). All shared peptides between human and mouse proteins were excluded for further analysis (bottom). In the final decision, only proteins with two or more peptides were counted. (i) it was determined that only human protein was identified. (ii) both protein determined to be unidentified. (**B**) The number of the proteins identified after applying the criteria in the analyses of single shot or high pH reversed phase fractions. With fractionation, the number of the identified human proteins increased approximately four times. (**C**) Secretory pathway analysis of the identified proteins with or without fractionation.

**Figure 5 ijms-23-06527-f005:**
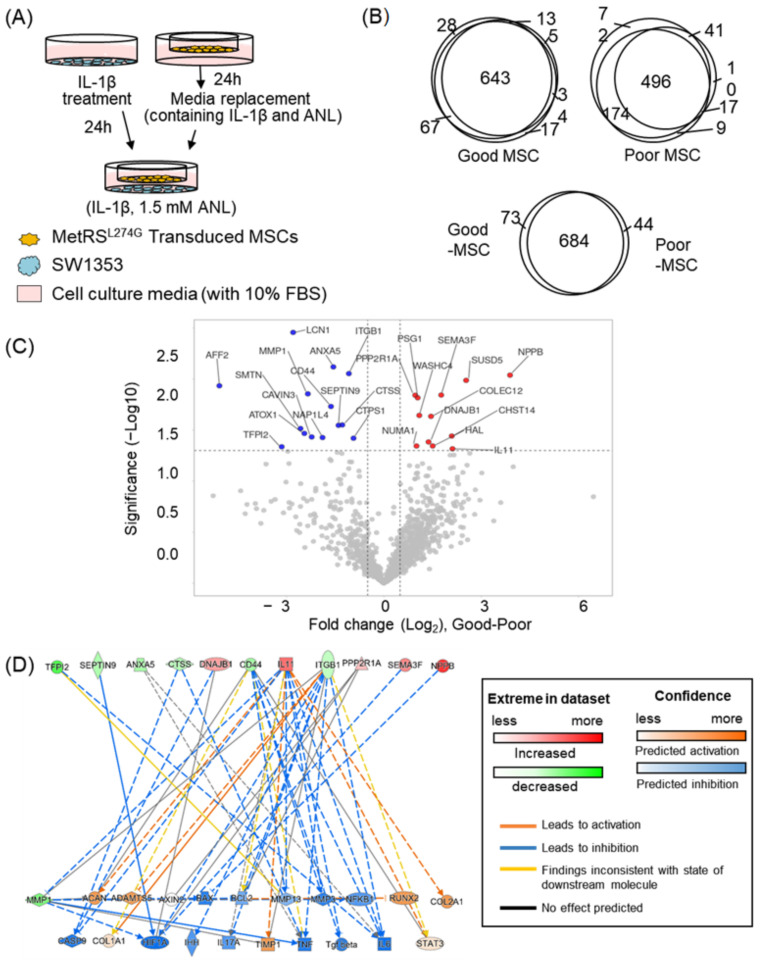
Analysis of secretome from MSCs co-cultured with osteoarthritis (OA)-induced cells. (**A**) Schematic workflow for analyzing secretome of MSCs that were co-cultured with OA-induced SW1353 cells. IL-1β was treated to induce OA. (**B**) The number of proteins identified in secretome of three good MSCs and three poor MSCs, as well as their inter- or intra-overlap; 90.4% of the identified proteins in the secretome of good MSCs were also identified in the secretome of poor MSCs. (**C**) Volcano plot displaying −log10 (*p*-value) vs. log2 (fold change, good MSCs over poor MSCs) for the identified proteins in the secretome of MSCs. Daggers indicate differentially expressed proteins between the two MSCs groups. Red and blue dots indicate proteins with increased expressions and decreased expressions in good MSCs, respectively. (**D**) Prediction of expression levels of OA-related genes based on the secretome profile using the Ingenuity pathway analysis (IPA) tool. The genes identified in our secretome profile are colored either green or red, while the genes whose expression levels are predicted by IPA are colored either blue or orange.

**Figure 6 ijms-23-06527-f006:**
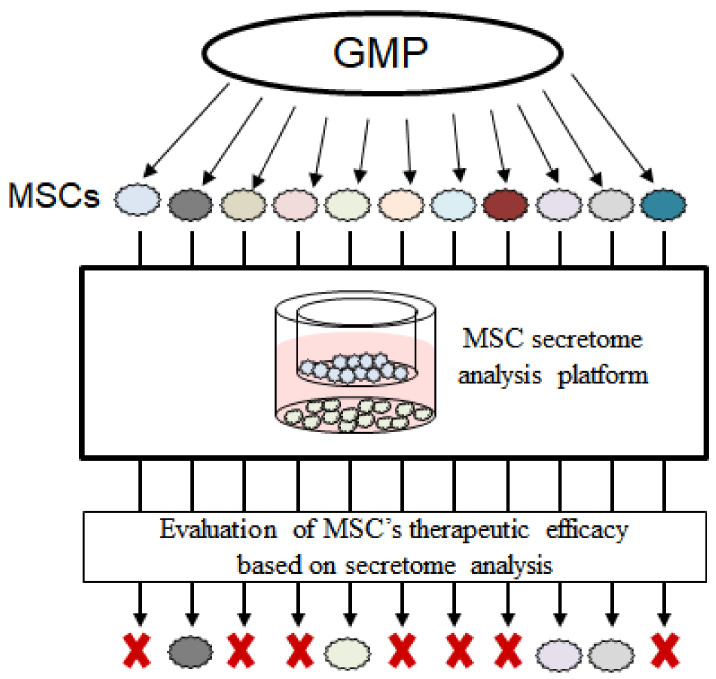
Application of MSC secretome analysis platform to evaluate the therapeutic efficacy of MSCs. Based on the secretome profile, MCSs with a better therapeutic effect could be determined and used to treat patients. Cross marks indicate that the MSCs are considered to have poor efficacy, based on secretome analysis.

## Data Availability

Not applicable.

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
