# Peer review of "Characterization of the Secretome of a Specific Cell Expressing Mutant Methionyl-tRNA Synthetase in Co-Culture Using Click Chemistry"

_ijms, 2022, doi:10.3390/ijms23126527_

Round 1
Reviewer 1 Report
The work was very well done in terms of content.
Minor language corrections needed.
More diagrams of these cellular models would be very useful to improve understanding of the work presented.
Author Response
Response to Reviewer 1 Comments
The work was very well done in terms of content.
We are grateful for the reviewer’s positive evaluation of our work.
Point 1: Minor language corrections needed.
Response 1: We corrected typos (including typos pointed by the other reviewer) in the revised manuscript, and the native English speaker proofread our manuscript.
Point 2: More diagrams of these cellular models would be very useful to improve understanding of the work presented.
Response 2: We appreciate this comment, and as the reviewer suggested, we added a diagram for improving the understanding of our work in the revised manuscript (Figure 6).
Reviewer 2 Report
This paper is torture to read. The organization and presentation detract from the science.
Parts of the Introduction belong in Results. Parts of the Discussion belong in Results. There does not appear to be a strong conclusion to the paper. The paper must be re-written hopefully with the help of an interested and experienced scientist with expertise in this area.
This reviewer cannot tell whether any strong conclusions resulted from the secretome proteomic analysis. This appeared to be the major thrust of the paper, but, there were few indications that analysis of the MSC secretome yielded significant insight into OA (osteoarthritis) or treatment.
What kind of cells are OA cells or corresponding “normal cells” (which require better description)? What is their relationship to MSCs?
Which of the proteins in Table I are secreted proteins by MSCs? What is the evidence? It appears that there is the secretome proteomic analysis and secretion targeting sequence analysis. Also, there is PCR gene expression data. It would be good to present all of the relevant characteristics of proteins in Table I, so the reader can make sense out of the presentation.
In Results, the authors must give a reason for each presentation, analysis and figure. Often, the reason for the analysis is not clearly stated and the conclusion from the analysis is not clearly stated (in Results).
Have experiments been done using secreted proteins with fluorescent tags to prove that these proteins are secreted? This should be explained.
Does a clear model emerge for how MSCs might ameliorate OA? This should be presented in Discussion. Remove explanation of the BONCAT strategy in Discussion. This paper is not about BONCAT technology.
The authors should not use “since” to mean “because”. Use “since” to indicate the passage of time.
Figure 1 Legend. “posttranscriptional regulatory elemental”. Should be “element”?
Please explain “click chemistry”.
Figure 5D. “Effect no predicted” is strange. Is “No effect predicted” better?
Good MSCs: either COL2A1 or STAT3 increased. This appears to be a weak conclusion.
Table I appears relevant. There is confusion about what MSCs secrete and what OA and normal cells express. Which cell is which?
Line 450: is “OA differentiation” good or bad? The meaning is unclear. Should this be “OA-pathology”?
Reviewer 3 Report
I consider that the article by Sungho Shin and collaborators is very interesting and it is a starting point for new research to find a therapy for osteoarthritis.
The article is well done, but I would suggest the authors make small additions.
It is necessary to add the country of origin of the equipment manufacturers.
I also think that the authors should also introduce the Conclusions section, given that the Discussion section is very complex.
In conclusion, I agree with the publication of the article.
Author Response
I consider that the article by Sungho Shin and collaborators is very interesting and it is a starting point for new research to find a therapy for osteoarthritis.
The article is well done, but I would suggest the authors make small additions.
We are grateful for the reviewer's positive evaluation of our work.
Point 1: It is necessary to add the country of origin of the equipment manufacturers.
Response 1: We added information about the country of origin in the revised manuscript.
Point 2: I also think that the authors should also introduce the Conclusions section, given that the Discussion section is very complex.
Response 2: We appreciate this suggestion, and as the reviewer suggested, we added the Conclusion section in the revised manuscript and revised the Discussion section to be less complex.
In conclusion, I agree with the publication of the article.
Round 2
Reviewer 2 Report
No further suggestions.
The paper is improved.
Author Response
Point 1: No further suggestions. The paper is improved.
Response 1: We are grateful for the comments raised by the reviewer. This has given us the chance to improve our manuscript. Once again, we thank the reviewer for the valuable comments.